# Haulout Patterns of Harbour Seal Colonies in the Norwegian Skagerrak, as Monitored through Time-Lapse Camera Surveys

**Elsa van Meurs** [1,*], **Even Moland** [1,2], **Arne Bjørge** [2] **and Carla Freitas** [2,3,*]

1 Centre for Coastal Research (CCR), University of Agder, 4604 Kristiansand, Norway; even.moland@hi.no
2 Institute of Marine Research, 4817 His, Norway; arne.bjoerge@hi.no
3 Marine and Environmental Sciences Center (MARE), 9020-105 Funchal, Madeira, Portugal
* Correspondence: elsavanmeurs@gmail.com (E.v.M.); carla.freitas.brandt@hi.no (C.F.)

**Abstract:** Harbour seals (*Phoca vitulina*) are part of the Norwegian coastal ecosystem and can be observed on skerries, islands, and sandbanks along the coastline, sometimes in close proximity to inhabited areas. In this study, we used time-lapse camera surveys to monitor the haulout patterns of harbour seals at two selected sites in the Norwegian Skagerrak, Lyngør and Østre Bolæren, over 12 and 4 months, respectively. The goal was to investigate how the number of seals hauling out on land varied seasonally and how it was influenced by environmental parameters (wind speed, air temperature, and water level), the time of the day, and anthropogenic disturbances. As expected, the number of seals hauled out increased with increasing air temperature and decreased with increasing wind speed and water level. Clear circadian patterns in the seal haulout behaviour were identified during autumn and winter when a significantly higher number of seals were observed on land at night. Moreover, haulout patterns showed significant seasonal variation, with a peak in haul outs being observed during the moulting season in August. Despite an expected high usage of land during the breeding season in early summer, the number of seals hauled out at the Lyngor study site was low during this period, especially during weekends and summer holidays, maybe due to increased disturbance from boats. This study provides valuable insights into the factors influencing the haulout behaviour of the species in the region and suggests possible effects of human disturbance on harbour seal behaviour in the area.

**Keywords:** haulout behaviour; pinnipeds; southern Norway; trail cameras; visual surveys





## 1. Introduction

Harbour seals (*Phoca vitulina*) and other pinnipeds are semi-aquatic mammals that have adapted to a diverse environment, making them able to live on land, on ice, and in water [1]. Harbour seals are relatively small seals and can live up to 22 to 35 years [2]. Adults are approximately 1.5 m and 70–100 kg [2]. Harbour seals are generalists, making them able to live off a wide array of food [3]. Seals living in the Skagerrak region primarily feed on smaller to medium-sized prey such as cod-related fish, sprat, and herring, as well as flatfish [4,5].

An adult harbour seal (aged one and older) goes through a shedding period, which is known as moulting [6,7]. The moulting process takes place in late summer, from August to September [8]. In this period, they struggle with thermoregulation and thus need to stay on land more. Harbour seals have also developed a behaviour that shortens the time used for moulting due to an external increase in skin temperature [9]. They haul out more frequently, enabling them to shorten their moulting period [8]. Additionally, they experience reduced heat loss on land compared to being in water.

Harbour seals are among the pinnipeds with the widest distribution, adapted to a vast variety of different habitats [3]. In Skagerrak, they inhabit coastal areas and show strong signs of site fidelity, making them more vulnerable to local disturbances. The habitat

may vary due to different seasons, providing them with different available nutrients. They typically inhabit coastal waters, estuaries, and bays and may wander offshore to feed [5].

Harbour seals are dependent on patches of land or ice to rest, breed, moult, and nurse young [1]. They prefer to haul out on rocky shores, skerries, beaches, or man-made platforms [3]. The amount of time spent on land varies depending on the season [1]. Many pinnipeds prefer more undisturbed areas, but harbour seals may inhabit areas near human presence [1].

Information on haulout patterns is important to understand the biology of the species and to aid management and conservation decisions in the Skagerrak region. Haulout patterns during the moulting season are particularly important for aiding the correction of population surveys [10]. This is because harbour seals are counted during the moult when they spend much time on land near their breeding sites [1].

External factors such as environmental parameters have a high influence on the haulout behaviour of harbour seals. Seasonality has an impact on the number of seals hauling out. For example, Granquist and Hauksson (2016) showed that the number of seals hauling out in Iceland decreases during the winter months and increases during the summer months [8]. They also observed the negative impact of wind speed and the positive effects of air temperature on the number of seals hauling out [8,11,12].

There are different findings on the diel differences in haulout behaviour, with some finding that it has no effect on the haulout behaviour [8], while others say that they prefer to haul out at night [12,13]. Tides also affect haulout behaviour due to the increase or decrease in areas to haul out on or the quality of sites available. Small boat traffic as an external factor might also have a temporary effect on the haulout behaviour of harbour seals [14].

The total population of harbour seals in Norway is about 10,000 individuals [15,16]. These seals can be found along the whole mainland coast of Norway to the west coast of Svalbard [15]. Harbour seals living in the Skagerrak area haul out on rocky shores, skerries, or smaller islands and may congregate in groups of up to tens of individuals [15,17].

During the last decades, the development of biotelemetry devices has allowed researchers to track individual seals and study haulout patterns at the individual level. Although these devices provide valuable individual information (for instance, on haulout duration), data on a few tracked individuals may be biased by individual traits, such as sex, maturity state, and individual personality. Moreover, few telemetry studies cover the pupping and moulting seasons during the summer, as tags are usually glued to the seal fur after moulting and then fall off during the next moult or, most often, before [18]. This makes it harder to study haulout behaviour over a full year because of the moulting period in which they lose their tags.

Trail cameras, on the other hand, can capture the natural behaviour of the study species with minimum disturbance. These cameras are especially adequate for studying animals in remote areas and might represent a good alternative to monitor haulout patterns throughout the year [19]. There are different ways to set up a camera trap: time-lapse mode takes an image during a set time frame, while motion mode captures an image when something is moving within the sensor range. Time-lapse cameras are cost-effective in the way that they collect large amounts of data compared to visual surveys, where a higher personnel demand is needed, as well as a higher chance of disturbing wildlife. Using time-lapse camera surveys to monitor an animal or animals enables the study of animals over time and obtaining a more complete continuous image of their surroundings [20]. Several studies have used trail cameras to observe animal behaviour. For instance, time-lapse photography, as well as visual monitoring and aerial surveys, have been used to observe how grey seal (*Halichoerus grypus)* and harbour seal haulout behaviour was affected by the construction of a large Danish offshore wind farm [21]. Another study used trail cameras to record behavioural responses of harbour seals affected by controlled disturbance trials as well as provide a daily seal count at certain sites in Scotland [22].

Using time-lapse camera surveys, this study investigated the haulout patterns of two harbour seal colonies in the Norwegian Skagerrak. Specifically, we investigated how the

number of seals on land was affected by diel, seasonal, and environmental parameters, as well as by anthropogenic disturbance. This study covered almost a full year in one location and four months at the second location.

## 2. Materials and Methods

### 2.1. Study Sites

This study was conducted at two known haulout sites for harbour seals in the Norwegian Skagerrak: Lyngør and Østre Bolæren (Figure 1).

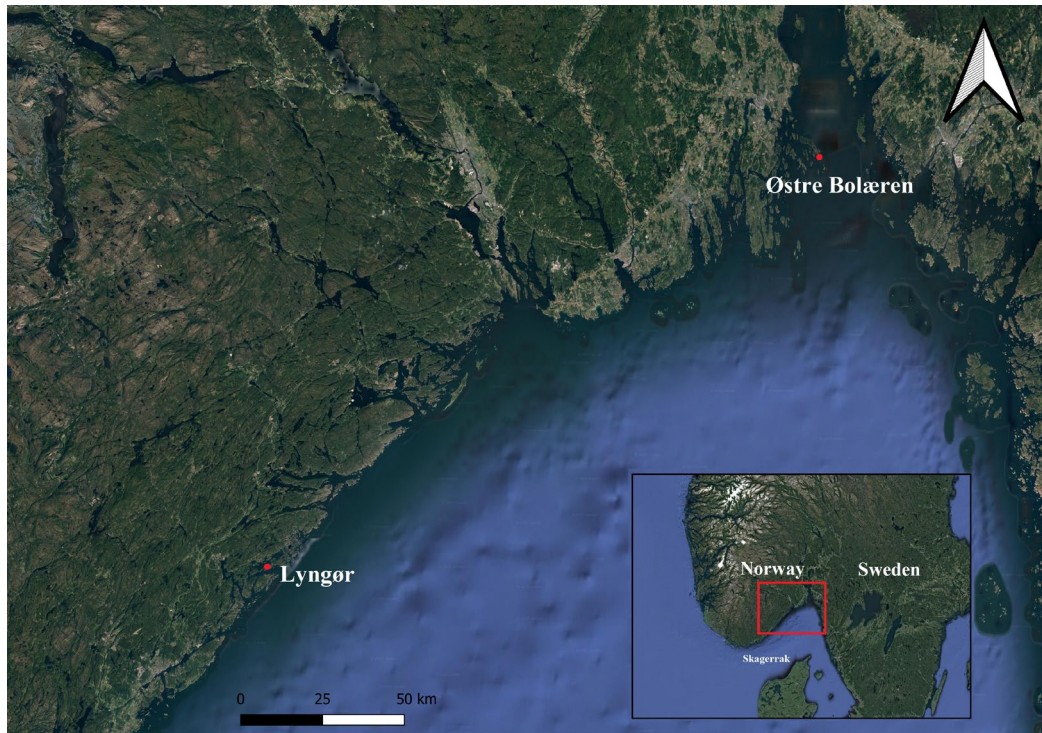

**Figure 1.** Map of the study area in the Norwegian Skagerrak showing the location of the two study sites: Lyngør and Østre Bolæren.

Lyngør (58.617602 N, 9.074661 E) is located in Tvedestrand municipality. The skerry on which the seals were located is surrounded by larger islands, which provide protection from wind and waves. Lyngør is a popular vacation spot during the summer and has high small-boat traffic as well as other water activities [23].

Østre Bolæren (59.192104 N, 10.583611 E) is located in Færder municipality and is part of Færder national park. The location is composed of several skerries and is exposed to wind and waves. The location is adjacent and in near proximity to the main shipping route to Oslo harbour.

### 2.2. Time-Lapse Camera Surveys

Three Bushwhacker Camo 4G time-lapse cameras, each with a SE5200 solar power kit to keep the batteries charged (Figure 2), were installed at the two study sites. The first camera was installed in February 2022 at Lyngør and the other two in September 2022 at Østre Bolæren. Cameras were equipped with an infrared night vision feature that reached approximately 20 m.

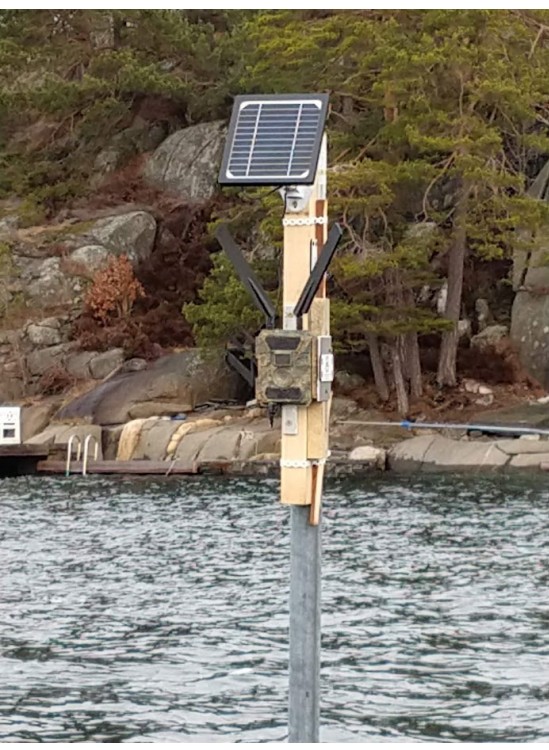

**Figure 2.** Time-lapse camera used in Lyngør. It consisted of a Bushwhacker Camo 4G camera connected to a solar panel for power supply. The same setup is used in Østre Bolæren. Photo credit: Elsa van Meurs.

Placing the time-lapse cameras in an adequate spot was a crucial part of this research. The camera needed to be placed in a location where it could capture most of the smaller and larger skerries. The camera also needed to be placed where people would not be able to block the field of view by standing in front of it. The distance from the seals also played an important part. The closer the camera was to the skerry, the easier it would be to differentiate seals from rocks. However, it was not possible to differentiate between age and sex. In Lyngør, there was only one main skerry where the seals could haul out. Østre Bolæren was covered with several skerries, making it more difficult to place only one time-lapse camera. Therefore, two time-lapse cameras were placed there, one facing south and another facing northwest, providing full coverage of the haulout sites. However, the south-facing camera failed after a few days and was therefore not used in the analysis. The northwest camera enabled capture of images at night because the camera was less than 20 m away from the observed animals and within range of the night vision feature. This camera was specifically used to test whether the number of seals varied significantly between day and night. The deployed time-lapse cameras took one image of the colony each hour. The images were sent via GSM cellular phone networks to an FTP site at the Institute of Marine Research, and each image automatically printed the date, time, air temperature, and moon phase.

*2.3. Environmental Data*

In order to investigate the effect of environmental conditions on the number of seals coming ashore, the following environmental variables were obtained for each hour: air temperature, wind speed, and water level. Air temperature (°C) was directly obtained from the time-lapse camera, which was calibrated before use. Wind speed (m/s) was obtained for each hour from Norwegian Centre for Climate Services [24]. The weather station Lyngør Fyr was used for Lyngør (which was 4.8 km in direct line from the study site), and Færder Fyr was chosen for Østre Bolæren (which was 14.5 km in direct line from the study site). Finally, water level was extracted from the Norwegian mapping authority [25].

*2.4. Data Analysis*

The number of seals in each image was manually counted and registered in Excel. In cases of adverse weather conditions where the seals were less visible but still reasonably distinguishable, count estimates were made. Otherwise, those images were excluded from the results. The camera at Østre Bolæren was able to capture seals at night, but only those laying within the first 20 m from the camera, not the individuals in the rocks further away. To obtain comparable data for day and night, only seals laying on the same rocks close by were counted during the night and day. Haulout behaviour, quantified by the number of seals on land, was examined in relation to environmental conditions (air temperature, wind speed, and sea water level) and temporal parameters (time of the year and time of the day). Additionally, anthropogenic disturbance on haulout patterns was examined by comparing the number of seals on land on working days versus weekends and holidays in Lyngør. Increased boat activity is expected to occur during weekends and holidays in this tourist destination, but was not directly quantified in this study, as time-lapse images at 1-h intervals were unlikely to reflect real boat activity in between images.. Summer holiday was defined as the period between 25 June and 14 August, which corresponded to school holidays. Weekends included both weekends (Saturday and Sunday) and public holidays except for the summer holiday, whereas workdays were all days Monday to Friday except for holidays. Statistical analyses were performed in R software (version 4.3.1). Data exploration was performed to locate outliers in the seal counts and environmental variables, as well as to check for correlations between explanatory variables to ensure that there were no collinearity issues. Outliers were identified by looking at data distribution and double-checked to see if the numbers were correct.

Generalised Linear Models (GLMs) with a Poisson distribution and log-link function were used to model the number of seals (response variable) as a function of the above-mentioned environmental and temporal parameters [26].

Two models were fitted, one for each locality. Both models used air temperature, wind speed, and water level as response variables. In Lyngør, where we had data for almost a full year, we also investigated seasonal variation in seal counts by adding month as explanatory variable. In addition, we investigated the eventual effect of recreational boat disturbance by including a variable type of day. The model for Lyngør, therefore, took the following form:

$$\log(\text{number of seals}) = \beta_0 + \beta_1 \text{ temperature} + \beta_2 \text{ water level} + \beta_3 \text{ wind} + \beta_4 \text{ type of day} + \beta_5 \text{ month}$$

For Østre Bolæren, the following model was used:

$$\log(\text{number of seals}) = \beta_0 + \beta_1 \text{ temperature} + \beta_2 \text{ water level} + \beta_3 \text{ wind} + \beta_4 \text{ diel}$$

In both models, number of seals (response variable) is the number of seals in each image, temperature is air temperature (°C), water level is water height above chart datum (cm), and wind is wind speed (m/s). Type of day is a categorical variable with three levels (working day, weekend, and summer holidays), month is calendar month, and diel is a categorical variable with 2 levels (day, i.e., solar elevation $\geq 0$, and night, i.e., solar elevation $< 0$). The $\beta_0$ coefficient is the intercept, i.e., the number of seals when all the predictor variables were equal to zero. Coefficients $\beta_1$ to $\beta_5$ represent the expected number in the log (number of seals) when each of the predictor variables increases with one unit (continuous variables) or when compared to the base level (categorical variables).

Models were fitted using the glm function in R software. Models were validated by calculating the dispersion parameter [27]. Both models were found to be overdispersed, as the dispersion parameters were greater than 1. To solve this issue, we fitted new models using a negative binomial distribution. Negative binomial models were fitted using the package MASS in R [28]. For Lyngør,

$$\text{Log(number of seals)} = \beta_0 + \beta_1 \times \text{temperature} + \beta_2 \times \text{water level} + \beta_3 \times \text{wind} + \beta_4 \times \text{type of day} + \beta_5 \times \text{month} + \log(\text{offset})$$

For Østre Bolæren,

Log (number of seals) = β0 + β1 × temperature + β2 × water level + β3 × wind + β4 × diel + log (offset)

The new model validation showed no overdispersion (Lyngør model: −0.58; Østre Bolæren model: 0.64).

## 3. Results

This study relies on data captured by time-lapse trail cameras positioned at two distinct locations within the Norwegian Skagerrak region. The time-lapse camera in Lyngør collected data for 12 months (17 February 2022–31 January 2023), with a single pause between 9 and 22 July due to a battery failure. A total of 8437 images were collected. In this period, the temperature ranged from −11.0 °C to 33.0 °C and had a mean temperature of 13.6 °C. Due to its wind-protected placement (Figure 3), the wind speed ranged from 0.0 m/s to 16.5 m/s and had a mean wind speed of 4.6 m/s. The water level ranged from −3.0 cm to 127.0 cm, with a mean water level of 53.3 cm. The number of seals on land ranged from 0 to 37, with an average of 9 (±7.6).

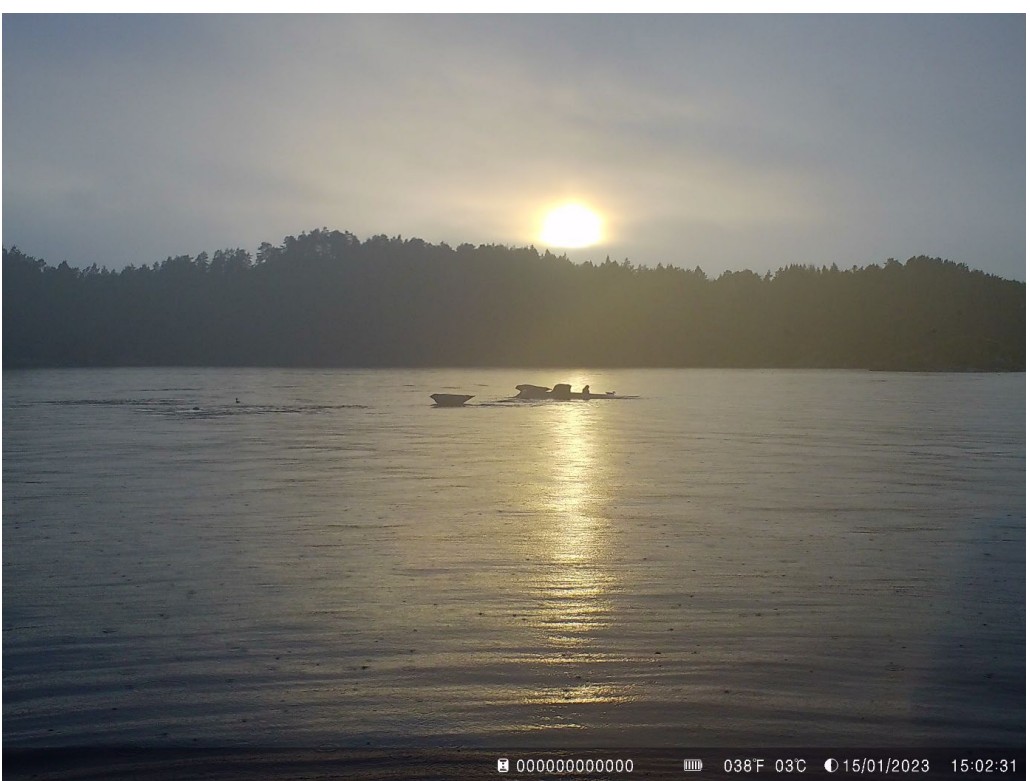

**Figure 3.** Example image from the time-lapse camera installed in Lyngør. This camera collected images between February 2022 and January 2023. Parameters registered by the camera are shown at the bottom of the image.

The camera in Østre Bolæren collected data for 85 days (29 September–22 December 2022), and a total number of 2022 images were obtained (Figure 4). Air temperature ranged from −4.0 °C to 26.0 °C, with a mean of 11.1 °C. Wind speed ranged from 0.3 m/s to 18.8 m/s (mean 8.8 m/s). Water level ranged from 1.0 cm to 130.0 cm and had a mean of 60.0 cm. The number of seals in the images ranged from 0 to 18, with an average of 1 (± 2.3) seal.

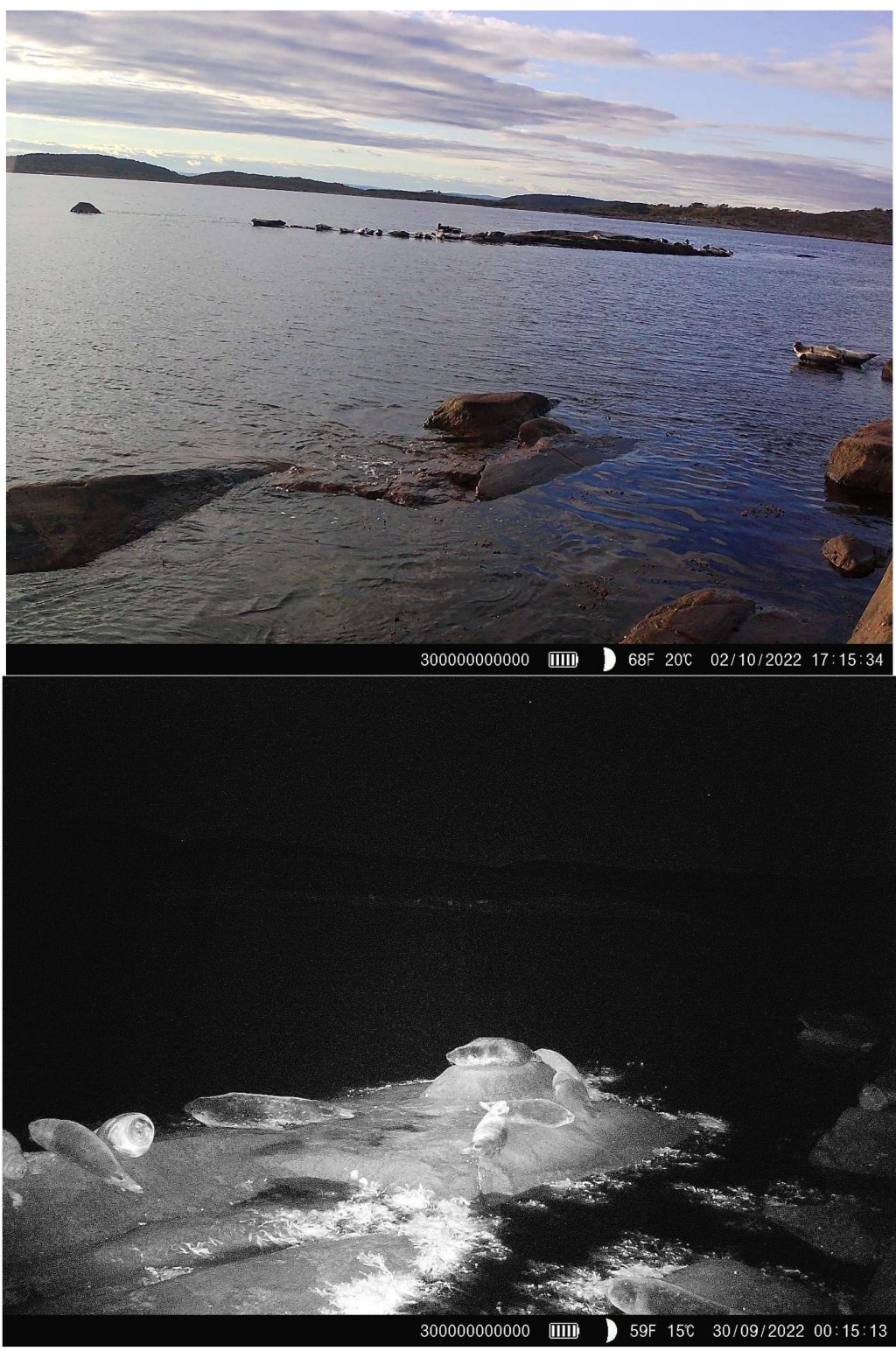

**Figure 4.** Example images from the time-lapse camera placed at Østre Bolæren obtained during the day (**above**) and night (**below**). Only seals on the nearest rocks (<20 m from the camera) were counted both during the night and day. This camera collected images between September and December 2022. Parameters registered by the camera are shown at the bottom of the image.

### 3.1. The Effect of Temperature, Wind, and Water Level

In both study sites, the number of seals hauled out increased significantly with air temperature (Tables 1 and 2). Data from Lyngør, which covered almost a year and a wide

range of temperatures, show that the rate of increase was slow at low temperatures, but it accelerated after the temperature reached approximately 25.0 °C (Figure 5).

**Table 1.** Model estimates and respective standard error (SE), Z-value, and *p*-value for the negative binomial Generalised Linear Models used to test the effect of wind speed (m/s), temperature (°C), water level (cm), month, and type of day on the number of seals (response variable) in Lyngør. Type of day was a categorical variable with three levels: working day (base level), weekend, and summer holidays.

| Parameter | Estimate | SE | Z | *p*-Value |
|---|---|---|---|---|
| (Intercept) | 3.1440692 | 0.0588757 | 53.402 | <0.001 |
| Wind speed | −0.1016158 | 0.0054498 | −18.646 | <0.001 |
| Temperature | 0.0097110 | 0.0018280 | 5.312 | <0.001 |
| Water level | −0.0155800 | 0.0008207 | −18.983 | <0.001 |
| Type of day (holidays) | −0.2217909 | 0.0455019 | −4.874 | <0.001 |
| Type of day (weekend) | 0.0100711 | 0.0346909 | 0.290 | 0.772 |
| Month | 0.0289606 | 0.0050355 | 5.751 | <0.001 |

**Table 2.** Model estimates and respective standard error (SE), Z-value, and *p*-value for the negative binomial Generalised Linear Model used to test the effect of wind speed (m/s), temperature (°C), water level (cm), and time of day (day versus night) on the number of seals (response variable) in Østre Bolæren.

| Parameter | Estimate | SE | Z | *p*-Value |
|---|---|---|---|---|
| (Intercept) | 2.58985 | 0.20812 | 12.444 | <0.001 |
| Wind speed | −0.08407 | 0.01487 | −5.654 | <0.001 |
| Temperature | 0.03997 | 0.01116 | 3.581 | <0.001 |
| Water level | −0.05564 | 0.00310 | −17.946 | <0.001 |
| Diel (night) | 0.83995 | 0.11969 | 7.018 | <0.001 |

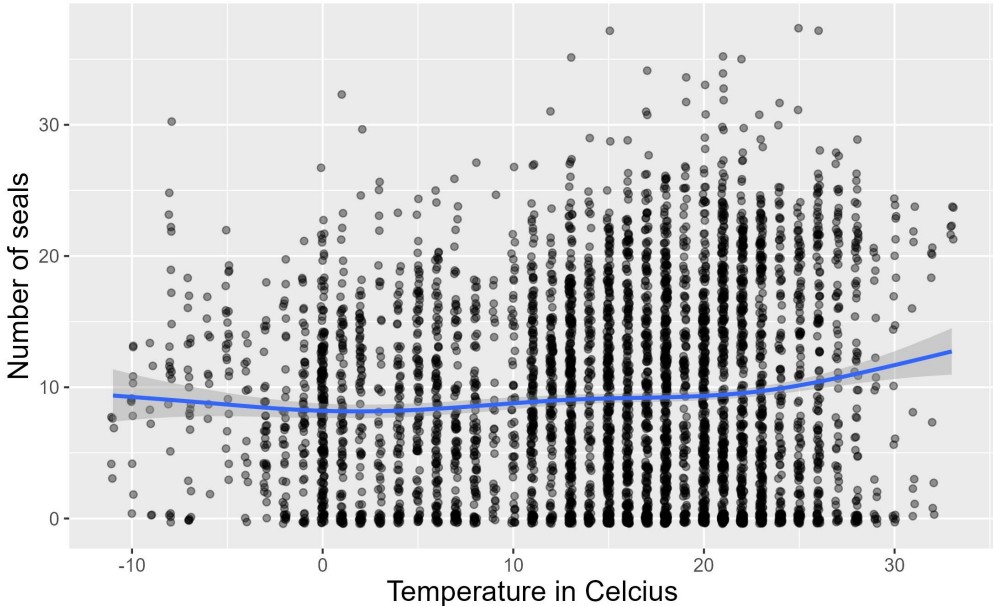

**Figure 5.** Relationship between the observed number of seals hauled out and the air temperature at Lyngør, Norway, in the period from 17 February 2022 to 31 January 2023. The smooth line shows the trend of the curve.

Conversely, wind speed and sea water level showed a negative effect on the number of seals on land in both study sites (Tables 1 and 2), as the number of seals hauled out decreased with increased wind speed (Figure 6) and with increased water level (Figure 7).

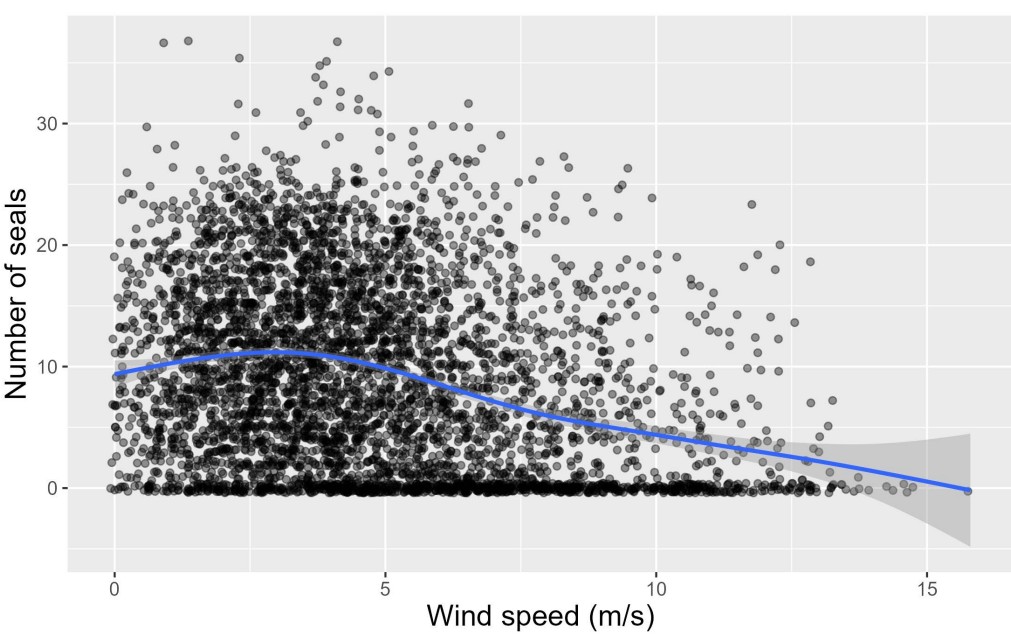

**Figure 6.** Relationship between the number of seals hauled out and wind speed at Lyngør, Norway, in the period from 17 February 2022 to 31 January 2023; the smooth line shows the trend of the curve.

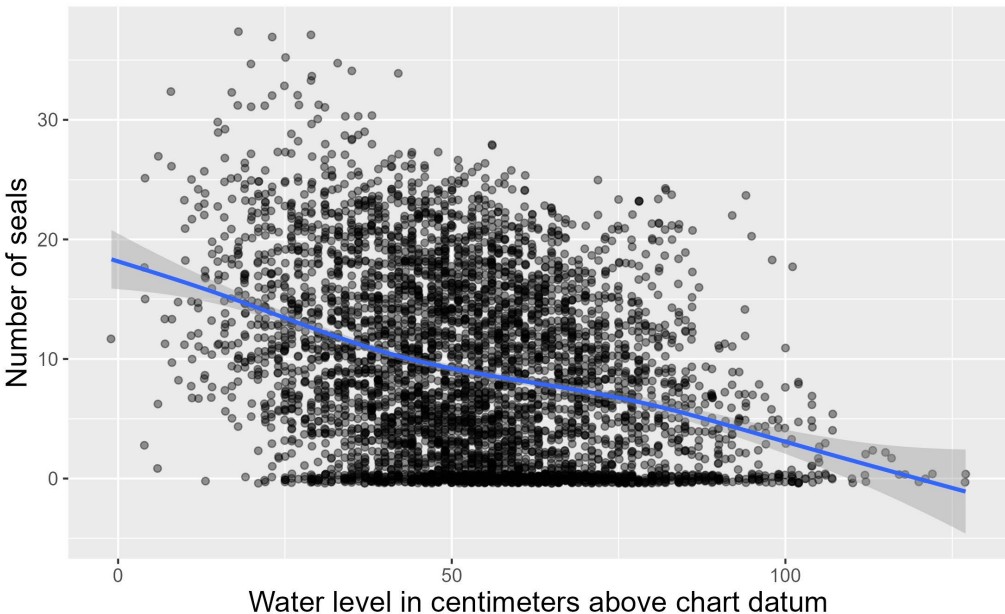

**Figure 7.** Relationship between the number of seals hauled out and the fluctuating water level in the tidal cycle at Lyngør, Norway, in the period from 17 February 2022 to 31 January 2023; the smooth line shows the trend of the curve.

*3.2. Seasonal Patterns*

Data from Lyngør, which covered almost a full year, show a seasonal variation in the number of seals on the haulout site. (Figure 8). Differences between months were statistically significant (Table 1). The average number of seals on land was generally larger in August and the beginning of September, which corresponds to the moulting period. However, the maximum number of seals (37) was observed in April. The lowest numbers were registered in June and beginning of July, which corresponds to the breeding season of the harbour seals. The number of seals also reached an overall low in seleted periods during winter, in association with unfavourable weather conditions.

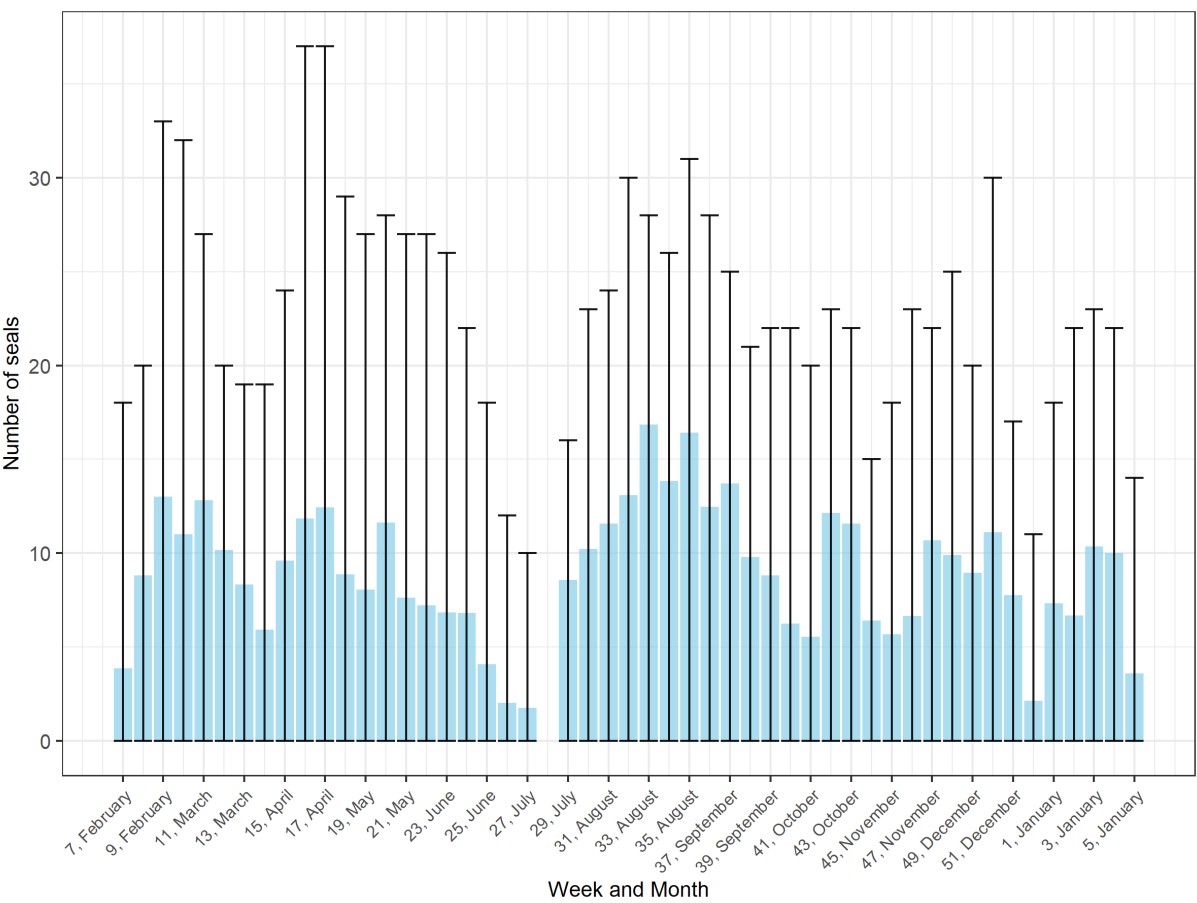

**Figure 8.** Average number of harbour seals per week in hourly images obtained from time-lapse camera installed in the Lyngør haulout site. Error bars indicate maximum and minimum number of seals observed during each week. The values missing during July (week 28 and part of week 29) were not incorporated in the means..

### 3.3. Human Disturbance, as Inferred from Holidays and Weekends

The possible effect of human disturbance on the number of seals on land was investigated by comparing seal numbers during workdays versus holidays and weekends. In general, the number of seals on land was lower during summer holidays and weekends during summer months (Figure 9). However, the same pattern was not observed during winter, and differences were not statistically significant (Table 1).

### 3.4. Diel Patterns

Images from Østre Bolæren allowed for the successful counting of seals hauling out during day and night in the period from 29 September to 22 December 2022. The sunrise in this period of time began at 07:17 at the end of September and ended at 09:12 on the 22 December. Sunset began at 18:57 at the end of September and ended at 15:19 on 22 December. In general, the number of seals on land was larger during the night than during the daytime (Figure 10), and differences were statistically significant (Table 2). The number of seals generally increased from 15.00 GMT (local wintertime −1), peaked in the period from 20.00 to 01.00, and decreased again towards dawn. In the period from 07.00 to 14.00, the number of seals decreased to the point that there were almost no seals to be spotted near the camera.

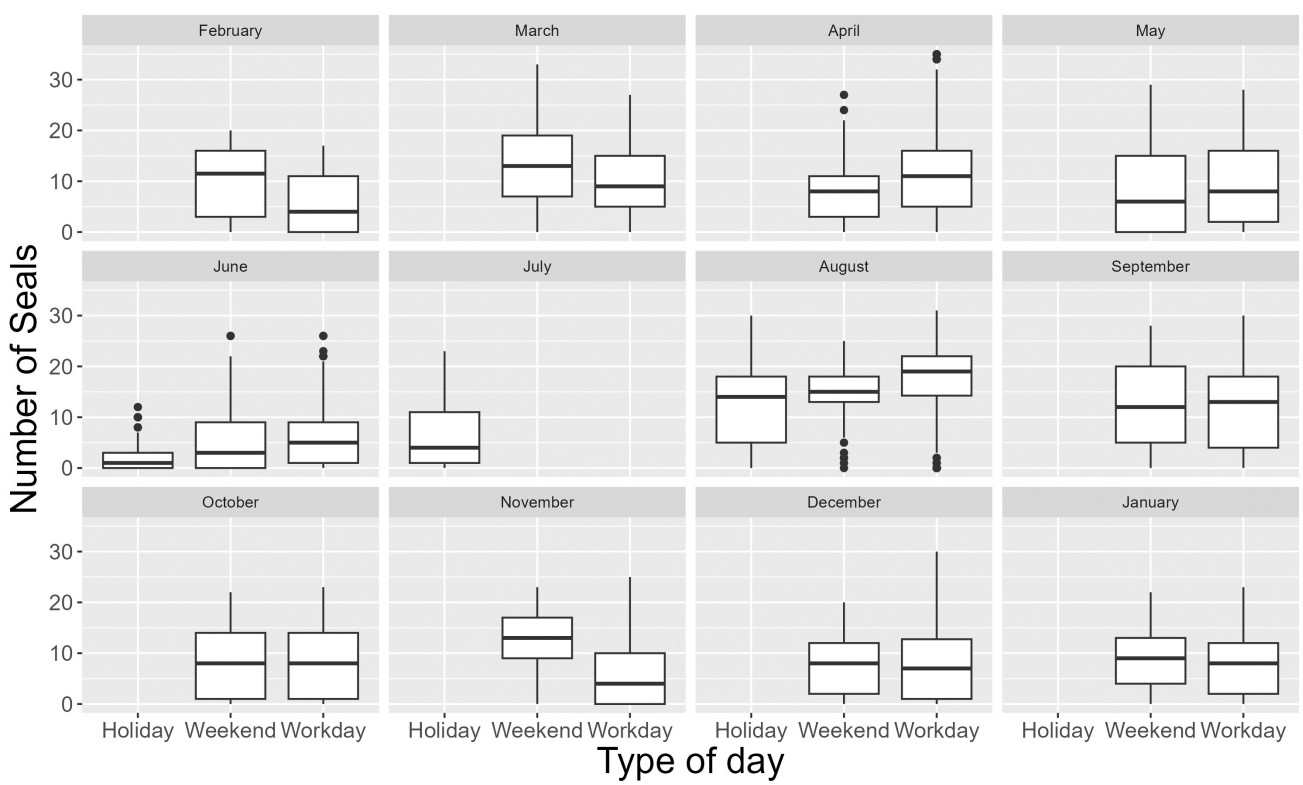

**Figure 9.** Boxplots showing the number of seals on land in summer holidays, weekends, and weekdays. Data for all months in 2022 were included, except for January, which was sampled in 2023. The "holiday" category encompasses the Norwegian summer holidays, which span from 27 June to 14 August. "Weekend" includes all Saturdays and Sundays (excluding those within the summer vacation period), as well as public holidays. Data were gathered in Lyngør between 17 February 2022 and 31 January 2023.

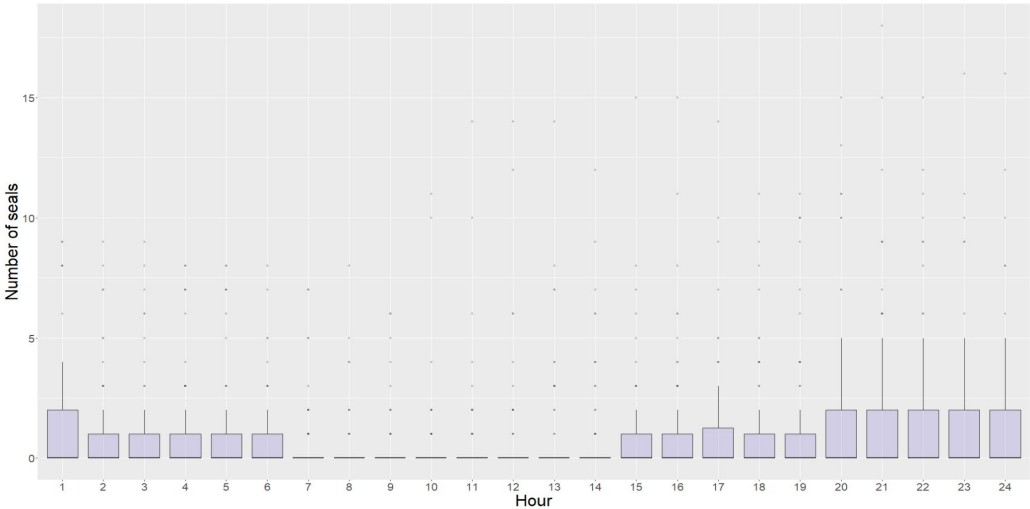

**Figure 10.** Boxplot showing the relationship between the observed number of seals on land in Østre Bolæren and time of the day. Hours are provided in GMT (local wintertime −1).

## 4. Discussion

Wind speed and water level showed a negative effect on haulout behaviour, while temperature had a positive effect. Moreover, the time of year, time of day, and the occurrence of weekends or holidays displayed differing effects on how the seals engaged in haulout

behaviour. These findings provide an insight into the harbour seal's haulout behaviour in the Skagerrak region—an area where seals inhabit coastal areas with a strong human presence.

Our study documents significant seasonal variation in the number of seals on land in Lyngør, a colony that was monitored for almost a year. The number of seals on land reached an overall peak in April, but was on average largest in August, with a mean number of 12 to 17 seals on land. This was not unexpected, as this period coincides with the moulting season for the species [29]. Conversely, an unexpectedly low number of seals on land was found in June and early July. This decrease may be associated with increased boat activity, as discussed further down.

After the moulting in August, there was a general decrease in the number of seals in the Lyngør haulout site. This could be attributed to seals not needing to haul out as much because of better thermoregulation than during the moult. They may also require an increase in body fat during the winter months; as temperatures drop, they rely on their body fat/blubber for insulation and might, therefore, spend more time in the water to feed [30].

Our study confirms previous findings regarding the effect of air temperature and wind speed on seal haulout behaviour. In general, there was a significant increase in the number of seals on land as the temperature rose. This might be explained by their thermoregulation and the benefits of coming ashore during warmer temperatures [31]. Being endothermic, these seals need to maintain a constant body temperature. They are able to regulate their heat loss, but because of the moult, their thermoregulation declines extremely, and therefore, they need to haul out more during warmer temperatures in order to maintain a constant body temperature [8,9]. Previous studies show that the relationship between temperature and haulout behaviour can be influenced by the distance to the foraging site. Seals that had longer foraging trips and could be at sea for several days displayed haulout behaviour less affected by temperature [32,33]. Wind speed had a negative effect on the number of seals hauling out, as increasing wind speed decreased the number of seals hauling out. Sea spray and waves due to strong wind exposure are also known to impact the haulout sites and decrease their quality for the seals [34]. Some studies performed on Weddell seals (*Leptonychotes weddellii*) have shown that seals tend to haul out less when the wind is strong, which, therefore, has a significant effect on the number of seals [35–37]. Calm wind, as well as warmer weather, generally promotes haulout behaviour [12].

Rising water levels decrease the dry areas available for seals to haul out on, thereby decreasing the number of seals hauling out. Some studies suggest a more direct connection between tidal level and haulout behaviour, hauling out more at the lowest tides [38]. However, there is variation in the influence of tides in different areas, with sites experiencing higher tidal influence being more impacted than those with lower tidal influence [34]. While some sites become available in low tide, other sites (e.g., steeper skerries) may become available only in high tide, but with an overall result where there are more haulout sites when the tides are low [8].

Human disturbance may affect local harbour seal use of haulout sites. During pupping season, the effect of human presence has a higher impact on the haulout behaviour as well as changing and impacting the pups [39,40]. Therefore, it is highly important to find out where the pups are and protect them from unnecessary human impact. Lyngør is, as mentioned, a highly attractive tourist location and has intense recreational boat traffic during the summer months. In this study, we found clear evidence that the number of seals decreased during the pupping season in June. Due to the battery issue in mid july, we may have missed the end of the pupping/weaning period. After the pupping season, seals may be less sensitive to boat disturbance, for instance during the moult in which they depend on obtaining good haulout sites to improve their thermoregulation [14].

It is possible that during the pupping season (June), seals move from the monitored skerry to the outer skerries where there is less boat activity. Indeed, no pupping activity has been observed in the monitored skerry during this study, while young pups have been

observed in the past in the outer skerries (personal observations). In Østre Bolæren, there is some seal-watching activity for tourists, as well as increased boat activity during summer. However, the colony was not monitored during this time of the year, so the potential effect of such activities could not be evaluated here.

The Østre Bolæren site offered a perfect opportunity to investigate diel patterns during the monitoring period in autumn and early winter. During that time, we found a clear circadian pattern in the haulout behaviour of harbour seals, with a preference for afternoons, nights, and early mornings and avoidance of midday haulout. This preference was also observed in earlier studies, where seals preferred hauling out during the afternoon and evening [12,41]. However, differences were noted between age, sex, and seasonal changes [12,41]. The reason why these harbour seals haul out during the darker hours might be because of several factors. By hauling out during the night, they may avoid human disturbance [42]. Previous studies have shown that diel patterns in haulout behaviour differ depending on the time of the year, which is affected by the prey they hunt [42]. It remains unclear whether the diel patterns observed in Østre Bolæren remain during spring and summer.

## 5. Conclusions

This study demonstrates the suitability of using time-lapse cameras to monitor the haulout patterns of coastal seals. Given regional differences between harbour seal populations in terms of abundance, distance to foraging areas, nature of haulout sites, and human disturbance, it is important to understand their haulout preferences in different localities in the Skagerrak region.

Our study confirms previous reports that the number of seals hauling out generally increases when temperatures rise and when wind speed decreases, as they benefit from coming ashore in warmer weather. Rising water levels due to changing tides and atmospheric pressure reduced the number of spots or made steeper areas available for seals to rest on, which in turn affected their haulout behaviour.

Harbour seals exhibited a clear circadian pattern during autumn–early winter, preferring to haul out during the night than during the day. This might be different in other localities and seasons—depending on disturbances and the suitability of haulout sites. Seal abundance surveys often involve counting seals during the moulting season. Further research on diel patterns during that season is particularly relevant for the eventual correction of count data. The impact of human disturbance on haulout behaviour, particularly during the reproductive season, may have a negative impact on seal populations. Time-lapse camera surveys using off-the-shelf commercially available trail cameras have the potential to further study such impacts on the behaviour of seals and other wild animals at a relatively low cost.

**Author Contributions:** Conceptualization, C.F.; methodology and fieldwork, E.M., C.F., and E.v.M.; formal analysis, E.v.M.; writing—original draft preparation, E.v.M.; writing—review and editing, C.F., E.M., and A.B.; supervision, C.F.; funding acquisition, C.F. and E.M. All authors have read and agreed to the published version of the manuscript.

**Funding:** This research received funding from the Institute of Marine Research and the University of Agder, Norway.

**Data Availability Statement:** The data presented in this study are available on request from the corresponding authors. The data are not publicly available due to ethical reasons related to possible seal disturbance and hunting.

**Acknowledgments:** We thank Færder National Park in Norway for permission to install the camera in Østre Bolæren. M. Biuw, M. Kristiansen, and M. Poltermann kindly helped with the acquisition and deployment of the camera in Østre Bolæren, and P.A. Boye helpfully collected the camera.

**Conflicts of Interest:** The authors declare no conflict of interest.

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
