# Peer review of "Haulout Patterns of Harbour Seal Colonies in the Norwegian Skagerrak, as Monitored through Time-Lapse Camera Surveys"

_diversity, doi:10.3390/d16010038_

Round 1
Reviewer 1 Report
Comments and Suggestions for Authors
Dear authors,
You are presenting a really interesting piece of work.
However, in my opinion, the way it is written, the strong assumptions made, the incorrect explanation of some of the results, and the incorrect use of specific references do not meet the scientific standards for publication.
I have included my comments by lines/paragraphs to clarify what I meant in the previous paragraph.
I do encourage you to improve the manuscript and resubmit it with better writing and references, as I believe it will be a valuable resource for the scientific community.
See my comments below/attached file.
Regards

Author Response
I am very grateful for all the comments I got on the paper and that you too the time to review it. I hope that you continue to review the article. I do have some simple questions to your comments. See them down below.
Comments 1: The authors are misspelling Lyngør
Response 1: I found no misspelled Lyngør in the abstract
Comments 2: The authors are explaining ‘the life history’ of the common seals in the area. However, I found it difficult to understand why, if their study and results are not linked with sex and maturity stage but rather focus on general patterns.
Response 2: Thank you for pointing this out, I changed the text and removed some pieces of the life history that is not relevant for this study.
Comments 3: The authors used a general reference when referring to common seal diet preferences. I strongly recommend that the authors use trophic ecology/diet studies on this species, as there are many of them in the north European waters.
Response 3: I found some articles that are more specific on the diet of Harbour seals in skagerrak
Comments 4: This sentence could be misleading to the reader. The authors indicated that seals feed on small and medium-size fish, which implies that they can consume prey ranging from 5cm in length (e.g. gobies) to 40 cm in length (e.g. Atlantic mackerel). Once again, incorporating a trophic ecology/diet study could provide a better understanding of what this pinniped regularly feeds on.
Response 4: I wrote down some examples of the organisms they eat that makes more sense than just writing down small to medium prey.
Comments 5: The authors are using information from a website provided by the Marine Institute Research. Such publications should not be used in scientific papers, particularly when there are several scientific publications related to this information. The authors should revise reference [7] and add a scientific publication.
Response 5: Observed that I used this reference a lot in the article and replaced it with other relevant articles instead.
Comments 6: The authors indicated that common seals are non-migratory species without providing any reference. Common seals are listed in Appendix II of the Bonn Convention (Convention on the Conservation of Migratory Species of Wild Animals). Despite the potential site fidelity, the species is considered migratory.
Response 6: Not sure if i totally agree on this, and found out that other articles don’t write down exactly if they are migratory or not, but I have removed it from the text.
Comments 7: The authors indicated that ‘the most determining factor for the harbour seal habitat is the type of food, its distribution, and its abundance’. The reference used in this statement has been picked up from a general one for pinnipeds. In L35, the authors mentioned that common seals are generalist species; therefore, they do not restrict their habitat to specific prey but rely on availability of prey. This has already been explained in several scientific publications on dietary patterns and trophic ecology of the species across its distribution area. This sentence is incorrect and should be removed. It is also in contradiction with the next paragraph (L55-L58)
Response 7: Thanks, I removed this sentence from the text
Comments 8: The reference does not have the correct format and should be considered reference [11]
Response 8: Changed it
Comments 9: The authors indicated that the relationship between the quality of haulouts and winds has been demonstrated in other studies. However, they only indicated a report, and they should add more studies justifying this
Response 9: Added more studies
Comments 10: should add more studies justifying this L74 Authors should remove the ‘,’ before [11]
Response 10: Changed it
Comments 11: It seems that the authors were starting a sentences with ‘Other studies..’ but a ‘.’ Is missing before ‘Other’
Response 11: Changed it
Comments 12: The reference used by the authors is a Master Thesis. There are several studies reporting this in a scientific publication document (e.g., https://doi.org/10.2307/2404296). Authors should use proper publications.
Response 12: Removed the master thesis from the paper
Comments 13: The authors should start a new paragraph with trail cameras and make both paragraphs flow. It appears that the mention of the latest technology (trail cameras) is merely inserted after the discussion of biotelemetry devices without a smooth transition, possibly because it needs to be included due to its relevance to their research.
Response 13: Started a new paragraph
Comments 14: The authors mentioned studies using the trail cameras on other vertebrates with a different behaviour. There are many studies on seals using such technology (e.g. 10.1371/journal.pone.0125486; https://www.nature.scot/sites/default/files/2017- 07/Publication%202015%20-%20SNH%20Commissioned%20Report%20894%20- %20Harbour%20seal%20haul-out%20monitoring%2C%20Sound%20of%20Islay.pdf; https://www2.dmu.dk/pub/sia_phd_web.pdf). It is recommended to use studies on species which their behaviour is similar to the ones authors are studying.
Response 14: Changed the paragraph and added the study
Comments 15: Both sentences should flow better. It seems that the last sentence was included to add the reference itself.
Response 15: Made them flow better, I hope
Comments 16: Because the authors included the website in the reference section, the website link should not be added here.
Response 16: removed the reference
Comments 17: The authors indicated that they examined the anthropogenic disturbance comparing among working days – weekends/holidays. At the paragraph L124-L128 they indicated that Lyngør might be a busy tourist area, however they do not indicate a similar situation for the Østre Bolæren location. Authors should explain how and why they do their comparisons with a better explanation.
Response 17: explained it better
Comments 18: Reference [26] should not be here, but at the end of the L204
Response 18: moved the reference
Comments 19: Authors should include a small paragraph explaining at what time the sunrise and the sunset/dawn occurs. It is difficult to understand that seals are more regularly haul-out from sunset to sunrise when the reader does not know this information from the study area. Authors should include a brief paragraph explaining the timing of sunrise, sunset, and dawn. It is challenging to comprehend that seals are more regularly haul-out from sunset to sunrise when this information, specific to the study area, it is not provided to the reader.
Response 19: explained it under the result section
Comments 20: I do recommend to authors to do not say “entire year” as the camera failed to record 9th22th July. It is not few days, but 14 days no working is almost half month. The recommendation is to say almost the entire year.
Response 20: changed entire year
Comments 21: I do recommend to authors maintain the same format for both equations. The equation used at the first site is clear, but the one used at the second site is more difficult to see
Response 21: changed the format
Comments 22: The authors should be careful when using decimals in measurements. If they say 13,6ºC they should say also -11,0ºC and 33,0ºC. If they say 53.3cm, they also should say -3,0cm and 127,0cm. I also do suggest to authors to say “0 to 37 individuals, with an average of 9 (±7,6) individuals”
Response 22: changed it
Comments 23: As mentioned for L234-L239 authors should be consistent when writing measurements.
Response 23: changed it
Comments 24: As mentioned at L206-L207 I do recommend authors to do not use the world ‘entire’ as it is not a precise word
Response 24: replaced the word
Comments 25: As mentioned at L206-L207 I do recommend authors to do not use the world ‘entire’ as it is not a precise word
Response 25: replaced the word
Comments 26: The authors indicated that the number of seals at the haulouts was larger in August, however, figure 8 shows that beginning of September also have similar numbers to other weeks in August. Also, they indicated that the lowest numbers were in June, but the graph shows that the lowest numbers are at the end of April-Beginning of May and end of December-beginning of January. The authors did not explain the gap in July in this section.
Response 26: I rewrote it and explained the gap in July
Comments 27: The authors indicated that the compared number of seals at the weekends in summer and winter between seasons and with workdays. However, this is not clear in the methodology. Authors should explain in the methodology this.
Response 27: Hope i made it more clear to the readers
Comments 28: The authors should specify the period of the year studied at this haul-out site. Additionally, the way they express the time schedule is hard to understand. I suggest that the authors use phrases such as 'between 8 pm to 1 am' and so on.
Response 28: don’t understand why it is hard to understand
Comments 29: These three sentences are results and they should not be included in this section.
Response 29: placed them under results
Comments 31: The authors should use references for the statement ‘Contrary to haulout studies-based individual-based tracking of seal behaviour’
Response 31: used a reference
Comments 32: As mentioned at L206-L207 I do recommend authors to do not use the world ‘entire’ as it is not a precise word
Response 32: changed it
Comments 33: The authors are using information from a website provided by the Marine Institute Research. Such publications should not be used in scientific papers, particularly when there are several scientific publications related to this information. The authors should revise reference [7] and add a scientific publication.
Response 33: added a scientific publication
Comments 34: The authors should make clear that this paragraph is related to the Lyngør haulout. They indicated in the middle of the paragraph misleading the reader.
Response 34: made it more clear
Comments 35: The authors indicated that the number of seals at the haulouts was larger in August, however, figure 8 shows that beginning of September also have similar numbers to other weeks in August. Also, they indicated that the lowest numbers were in June, but the graph shows that the lowest numbers are at the end of April-Beginning of May and end of December-beginning of January.
Response 35: changed it
Comments 36: This is not an study on seal condition and how seals thermoregulate, therefore I do recommend to authors to use other particles (such could, should, might, may) instead of the use of ‘can’.
Response 36: changed it
Comments 37: Reference [4] is a general reference and authors should use a more specific reference regarding this topic. The Kince at al reference might have references rakated to their heat loss.
Response 37: used a more specific reference
Comments 38: Reference [31] is a Master thesis. Authors should use a peer-review publications instead this kind of publications
Response 38: removed the reference
Comments 39: Common seals usually foraged in areas within the first 30km from shore (https://www.abdn.ac.uk/sbs/documents/tollit98.pdf). In theory foraging sites might be relatively close to their haulout sites.
Response 39: articles say differently
Comments 40: The authors indicated at the beginning that seals are non-migrant species. While they are migrant species, this part of the text is contrary with what the mentioned in the Introduction section.
Response 40: not quiet sure if everyone agrees on that
Comments 41: The authors are indicated that some sites are affected by tides, giving the sample of the Østre Bolæren haulout. However, the study at that site was only performed between September and December
Response 41: that doesn’t mean that it cannot be affected by tides…
Comments 42: The authors indicated that the pupping season is June-July and seals move out from the Lyngør haulout. However, the camera was not working from 9th-22nd July, and they should mention that some biases might occur due to this circumstance.
Response 42: mentioned that there were some biases
Comments 43: The authors indicated that other studies have shown more haulout activity at night, but they only mentioned one scientific study. The authors should include more.
Response 43: included more
Comments 44: The authors are assuming than seals are foraging only at daytime but nor reference was included. Seals can feed also at night if they find a source of light (e.g., fishing boat). The authors should include references justifying these statements.
Response 44: hope I justified it better
Comments 45: The authors are assuming that the foraging areas for seals are far away from their haulouts, however seals are usually feeding within the 30km from their haulouts. Authors should be careful when doing these statements
Response 45: not quiet sure
Comments 46: The authors should write the references correctly
Response 46: don’t know what is wrong with them
Comments 47: The authors should specify the Chapter in the book they meant
Response 47: specified it
Comments 48: The authors are using information from a website provided by the Marine Institute Research. Such publications should not be used in scientific papers, particularly when there are several scientific publications related to this information. The authors should revise reference
Response 48: changed it
Comments 49: Format of the legend should be amended, a square with the contents is visible.
Response 49: changed it

Reviewer 2 Report
Comments and Suggestions for Authors
This study builds upon the work of others by using a relatively new(ish) technology to study phenological patterns of harbor seals in Norway. This is an important contribution to the literature, because it has critical implications for the design of other scientific studies (notably, how do we conduct population censuses when we know we aren't capturing the entire population), as well as for environmental managers (e.g. in the context of human-wildlife conflict). I applaud the authors for their work reporting on natural history observations that are often overlooked in the literature but have important implications.
With that being said, the manuscript needs substantial improvement prior to publication. I have attached a PDF with inline comments that I hope will be useful to the authors in strengthening the manuscript, analyses, and conclusions.

Comments on the Quality of English LanguageOverall, the authors wrote a comprehensible manuscript. There were several issues throughout the manuscript, but appeared to be typos and editing issues than anything relating to the quality of the English. I would suggest that the authors carefully review the manuscript for typos and incomplete sentences.
Author Response
I am very grateful for all the comments I got on the paper and that you too the time to review it. I hope that you continue to review the article. I really appreciated that you made the comments in the text and that you also looked at the grammatical issues. I have made changes to all your comments. A huge thanks for having the time to review my paper!